# Mitochondrial Dynamics and Mitophagy in Skeletal Muscle Health and Aging

**DOI:** 10.3390/ijms22158179

**Published:** 2021-07-30

**Authors:** Jean-Philippe Leduc-Gaudet, Sabah N. A. Hussain, Esther Barreiro, Gilles Gouspillou

**Affiliations:** 1Research Institute of the McGill University Health Centre, Department of Critical Care, Montréal, QC H4A 3J1, Canada; sabah.hussain@muhc.mcgill.ca (S.N.A.H.); gouspillou.gilles@uqam.ca (G.G.); 2Meakins-Christie Laboratories, Department of Medicine, McGill University, Montréal, QC H4A 3J1, Canada; 3Département des Sciences de l’activité Physique, Faculté des Sciences, UQAM, Montréal, QC H2X 1Y4, Canada; 4Pulmonology Department-Muscle Wasting & Cachexia in Chronic Respiratory Diseases & Lung Cancer Research Group, IMIM-Hospital del Mar, Parc de Salut Mar, Biomedical Research Park (PRBB), C/Dr. Aiguader, 88, 08003 Barcelona, Spain; ebarreiro@imim.es; 5Centro de Investigación en Red de Enfermedades Respiratorias (CIBERES), Instituto de Salud Carlos III (ISCIII), 28029 Madrid, Spain; 6Health and Experimental Sciences Department (CEXS), Pompeu Fabra University (UPF), Biomedical Research Park (PRBB), C/Dr. Aiguader, 88, 08003 Barcelona, Spain; 7Centre de Recherche de l’Institut Universitaire de Gériatrie de Montréal, Montréal, QC H3W 1W5, Canada

**Keywords:** skeletal muscle, mitochondrial dynamics, mitophagy, autophagy, aging, sarcopenia

## Abstract

The maintenance of mitochondrial integrity is critical for muscle health. Mitochondria, indeed, play vital roles in a wide range of cellular processes, including energy supply, Ca^2+^ homeostasis, retrograde signaling, cell death, and many others. All mitochondria-containing cells, including skeletal muscle cells, dispose of several pathways to maintain mitochondrial health, including mitochondrial biogenesis, mitochondrial-derived vesicles, mitochondrial dynamics (fusion and fission process shaping mitochondrial morphology), and mitophagy—the process in charge of the removal of mitochondria though autophagy. The loss of skeletal muscle mass (atrophy) is a major health problem worldwide, especially in older people. Currently, there is no treatment to counteract the progressive decline in skeletal muscle mass and strength that occurs with aging, a process termed sarcopenia. There is increasing data, including our own, suggesting that accumulation of dysfunctional mitochondria contributes to the development of sarcopenia. Impairments in mitochondrial dynamics and mitophagy were recently proposed to contribute to sarcopenia. This review summarizes the current state of knowledge on the role played by mitochondrial dynamics and mitophagy in skeletal muscle health and in the development of sarcopenia. We also highlight recent studies showing that enhancing mitophagy in skeletal muscle is a promising therapeutic target to prevent or even treat skeletal muscle dysfunction in the elderly.

## 1. Introduction

Skeletal muscle, which accounts for over 40% of total body mass, is highly adaptable and of critical importance for general health. Indeed, it plays key roles in posture, mobility, thermogenesis, glucose homeostasis, and energy metabolism. The regulation of skeletal muscle mass is mainly controlled by a fine balance between protein synthesis and protein degradation [1]. Skeletal muscle has a striking ability to adapt to various physiological and pathological conditions by changing its size, composition, and metabolic properties. Muscle atrophy, or the loss of muscle mass, is an important clinical problem that leads to negative health outcomes, especially during aging.

The progressive decline in muscle mass and strength that occurs with aging, a process known as sarcopenia, can have serious consequences on the lifestyle and functional status of the elderly. For instance, lower muscle mass and strength has been associated with functional impairment, falls, and physical disability [2,3]. Sarcopenia is a strong predictor of hospitalization and is highly linked to short- and long-term mortality in older adults [4,5,6]. Given the increasing population of older adults all over the world, it is not surprising that the age-related decline in muscle mass, strength, and function has become a major global health concern. Highlighting its serious health consequences, sarcopenia was recognized in 2016 as a muscle condition/disease and received its own ICD (international classification of diseases)-10-Clinical Modification code (M62.84) [7].

Several biological mechanisms have been proposed to explain the development of sarcopenia [8]. Among these mechanisms, strong experimental evidence indicates that the accumulation of mitochondrial dysfunction plays an important role in the muscle aging process. Indeed, many studies conducted in both rodents and humans have shown that aging is associated with a decline in mitochondrial bioenergetics [9,10,11]. Several studies have also shown that the mitochondrial permeability transition pore (mPTP) becomes dysfunctional in old rodents and humans [9,12,13,14]. Alterations in mPTP function are undoubtedly contributing to the aging-related loss of muscle mass and function, since mPTP opening can trigger several atrophy-regulating pathways. First, mPTP opening can result in mitochondrial reactive oxygen species (ROS) overproduction [15]. ROS can, in turn, trigger the muscle atrophy program by activating the Fork head box O (FoxO) transcription factor family [16,17]. Second, mPTP opening can result in mitochondrial DNA release and consequential activation of the NLRP3 (NOD-, LRR- and pyrin domain-containing protein 3) inflammasome, which was shown to increase the expression of ubiquitin ligases involved in muscle atrophy [18,19]. Third, cytochrome C release secondary to mPTP opening was shown to increase proteasomal activity secondary to caspase 3 activation [20]. Fourth, mPTP opening can lead to the release of pro-apoptotic factors such as endonuclease G, which was shown to accumulate in myonuclei of old individuals [9]. Further strengthening the view that mitochondria play key roles in the aging-related loss of muscle mass and function, the most efficient interventions to attenuate sarcopenia are exercise training [21,22,23,24] and calorie restriction [25,26,27], both well-known to improve mitochondrial health [28].

While the accumulation of mitochondrial dysfunction is now generally seen as a hallmark of muscle aging, the mechanisms underlying the progressive decline in mitochondrial function remain poorly understood. In this article, we critically review the available literature that has positioned impairments in mitochondrial dynamics and mitophagy as potential contributors. We also review recent evidence showing that targeting mitophagy might be a promising strategy to prevent or even treat sarcopenia.

## 2. The Importance of Mitochondria Dynamics for Skeletal Muscle Health

Initially considered as bean-shaped organelles, mitochondria are now known to display a complex architecture, especially in skeletal muscle cells [29]. Mitochondria are dynamic organelles that are constantly undergoing fission and fusion so as to adapt their morphology to the cellular environment (Figure 1). Mitochondrial fusion is mainly regulated by the Mitofusin 1 & 2 (Mfn 1 & 2) and OPtic Atrophy 1 (Opa 1) proteins. Mitochondrial fission is mainly governed by Dynamin-Related Protein 1 (Drp1) (reviewed in [30,31]). Several other proteins targeting the outer membrane, such as mitochondrial fission 1 protein (Fis1), mitochondrial fission factor (Mff), and mitochondrial dynamic protein 49 and 51 (MiD49, and MiD51), participate in the fragmentation of mitochondria [32]. Figure 1 shows the known regulators involved in mitochondrial fusion and fission.

In the past decade, several studies have established the importance of mitochondrial dynamics in maintaining skeletal muscle and mitochondrial integrity and function. It was, for instance, reported that mice lacking both Mfn-1 and Mfn-2 in skeletal muscle display severe mitochondrial dysfunction, accumulation of mitochondrial DNA damage, and a severe deficit in growth [33]. Mfn2 deletion in skeletal muscle was shown to result in oxidative stress and muscle atrophy in adult mice [34]. Deletion of both Mfn1 and Mfn2 in adult skeletal muscle resulted in a major decrease in exercise performance, indicating that mitochondrial fusion is essential to muscle performance in mice [35].

Specific deletion of Opa1 in skeletal muscle leads to mitochondrial dysfunction, oxidative stress, endoplasmic reticulum (ER) stress, and inflammation [36,37,38]. Opa1 deficiency also promotes the secretion of Fibroblast Growth Factor 21 (FGF21) from skeletal muscle, which leads to altered lipid homeostasis, inflammation, and the senescence of different tissues [36]. Taken altogether, the available literature clearly indicates that mitochondrial fusion proteins are essential for the maintenance of mitochondrial and skeletal muscle health.

Several recent studies have highlighted the physiological importance of mitochondrial fission in the maintenance of skeletal muscle health. Muscle-specific Drp1 overexpression was shown to impair skeletal muscle growth in mice [39,40]. Inhibition of mitochondrial fission by genetic silencing of Fis1 and Drp1 in skeletal muscle has been shown to protect against muscle wasting induced by starvation or by overexpression of the atrophying transcription factor FoxO3a [41]. Muscle-specific Drp1 knockout in mice leads to a severe myopathic phenotype including muscle wasting, weakness, and signs of muscle degeneration and regeneration [42]. The myopathic phenotype was evident about 2 months after muscle deletion of Drp1 and was also present at 6 months [42]. The importance of Drp1 in regulating muscle function was further emphasized by our group [43], which revealed that muscle-selective Drp1 knockdown decreases mitochondrial respiration and increases markers of muscle regeneration, denervation, fibrosis, and oxidative stress. Interestingly, both Drp1 knockout and knockdown altered autophagy and mitophagy levels [42,43]. As mentioned above, overexpression of Drp1 impaired skeletal muscle growth [39], whereas deletion or insufficient expression of Drp1 in skeletal muscle led to 40–50% atrophy [42,43], indicating that Drp1 is a key player in regulating skeletal muscle development and maintenance. Evidence from *Drosophila* also indicates that manipulating mitochondrial dynamics can impact myofibril development [44]. Another recent study showed that loss of Fis1 leads to mitochondrial dysfunction, proteostasis impairment, muscle degeneration, and reduced lifespan in Fis1 mutant flies [45]. Taken altogether, the findings discussed above clearly indicate that mitochondrial fission is essential for skeletal muscle health.

## 3. Mitochondrial Dynamics in the Skeletal Muscle Aging Process: Current State of Knowledge and Considerations for Future Research

As highlighted above, the available literature clearly positions mitochondrial dynamics as key processes in the maintenance of muscle and mitochondrial health. It is, therefore, not surprising that many studies have investigated whether impairment in mitochondrial dynamics could contribute to sarcopenia. However, and as will be reviewed below, the involvement of mitochondrial dynamics in the skeletal muscle aging process remains unclear as conflicting data exist in the field (Table 1).

In their elegant study, Sebastian et al. reported that mice lacking Mfn2 in skeletal muscle from birth display accelerated sarcopenia [34]. They also reported that aged control (wild type, WT) mice displayed a reduction in Mfn2 content [34]. However, the finding that Mfn2 content declines with aging is far from being universal. Indeed, Mfn2 content has been reported as decreased [34,49,55], unchanged [52], or even increased in aged rodent skeletal muscles [47,50,54]. In humans, while Mfn2 expression was decreased in aged skeletal muscle in one study [60], others have reported either similar [56,58,59,61] or even higher Mfn2 content in aged skeletal muscles [61]. Smartly conflicting data also exist for Drp1 and Opa1.

In a recent study, Tezze et al. reported that Opa1 content normalized to complex II was lower in muscle from old sedentary individuals but not in old sportsmen. Interestingly, Opa1 content normalized to complex II content correlated with muscle fiber diameter and specific force in elderly individuals [36]. Somewhat in line with this finding, Joseph et al. reported that Opa1 content was lower in the skeletal muscle of both low functioning and high functioning individuals [59]. However, several studies did not detect any impact of aging on Opa1 content in the skeletal muscle of aged mice [52,54], rats [50], and humans [57,58]. It is worth mentioning that Tezze et al. also reported that 18-month-old mice display lower levels of Opa1, but also showed that one week of exercise training was sufficient to restore Opa1 in aged skeletal muscles [36].

Conflicting data also surround the impact of aging on Drp1 content. Indeed, studies in rodents have reported increased [49,50,51] or unchanged [34,47,52,54] Drp1 content in aged skeletal muscles. The literature in humans also contains conflicting data, with some studies reporting a decreased [58] or an unchanged [59] Drp1 content in skeletal muscle of old individuals.

Clearly, no consensus on the role that mitochondrial dynamics play in the muscle aging process has been reached. Recently developed transgenic mouse lines with photoactivatable mitochondria [62] might prove useful in addressing this issue. Defining whether impairment in mitochondrial dynamics is a likely contributor to the aging-related loss of muscle mass and strength in humans will require longitudinal studies. Such studies will need to pay particular attention to physical activity levels of participants, as the latter can impact the expression of proteins regulating mitochondrial dynamics [63,64,65]. Further studies will also need to take into account the possibility that the impact of aging on mitochondrial dynamics and morphology might be fiber type-specific. Indeed, we recently reported in aged rats that the oxidative soleus muscle displayed signs of mitochondrial fragmentation, while the glycolytic white gastrocnemius displayed signs of mitochondrial elongation with aging [50]. Other studies have also highlighted that mitochondrial morphology and dynamics differ across muscle fiber types [61,66,67]. Importantly, a major gap in knowledge in the field that needs to be addressed urgently is whether sex differences exist in the role that mitochondrial dynamics play in the skeletal muscle aging process. Indeed, and to the best of our knowledge, no previous study has investigated whether the content of proteins regulating mitochondrial dynamics is differently affected in aged males vs. females.

Studies aiming at defining whether mitochondrial dynamics can be targeted to counteract sarcopenia are also needed. To date, the modulation (overexpression or deletion) of proteins regulating mitochondrial dynamics in mice has either shown no benefit to muscle cells [68] or was proven deleterious to muscle health (Drp1 overexpression, Drp1 knockout, Drp1 knockdown, and Mfn1+Mfn2 knockout). Strengthening the need for such mechanistic studies, promoting Drp1-mediated mitochondrial fission in middle-aged flies was shown to extend lifespan and improve muscle mitochondrial function, likely by facilitating mitophagy [69] (mitochondrial fission can, indeed, facilitate the removal of damaged/dysfunctional mitochondria through mitophagy [70]). This study, therefore, suggested that Drp1 overexpression could be a potential strategy to attenuate the aging-related accumulation of mitochondrial dysfunction and counter sarcopenia. However, we recently reported that knocking down or overexpressing Drp1 late in life in mouse skeletal muscle (from 18 to 22 months of age) is detrimental to skeletal muscle mass and mitochondrial health [71]. These recent findings, therefore, indicate that targeting Drp1 expression is unlikely to be a viable target to counter sarcopenia and indicate that mitochondrial fission must be maintained within a specific physiological range to preserve muscle and mitochondrial health. Whether enhancing/overexpressing mitochondrial fusion proteins, such as Mfn2 in muscles of aged WT mice, can attenuate the progression of sarcopenia has never been tested.

## 4. The Role of Autophagy in Skeletal Muscle Health and Aging

The word autophagy derives from two Greek words “auto”, meaning self, and “phagy”, meaning eating. Three different delivery systems have been described in mammals: macro-autophagy, chaperone-mediated autophagy, and micro-autophagy. In this review, the term autophagy is synonymous with macroautophagy, which is characterized by the formation of double-membrane vesicles known as autophagosomes that surround portions of cytoplasm, organelles, and protein aggregates that are subsequently delivered to lysosomes for degradation. The proper maintenance of this catabolic process is essential for survival and has important roles in various physiological functions [72,73]. Under normal conditions, autophagy mainly prevents the accumulation of damaged organelles and misfolded proteins. In response to stress, such as starvation, autophagy acts mainly as a pro-survival mechanism, by providing metabolic substrates such as amino acids through the bulk degradation of cytoplasmic components; however, excessive self-degradation can lead to cell death and pathological phenotypes [74,75].

The formation of autophagosomes and the degradation of its cargo involve several steps including initiation, elongation, maturation, fusion, and degradation (Figure 2). These steps require energy and involve more than 30 autophagy-related (ATG) proteins in yeast, and many of these proteins are highly conserved in mammals [76,77].

The initiation and expansion of the phagophore depend on the activation of the ULK1 and PI3-K complexes and the synthesis of phosphatidylinositol 3-phosphate (PI3P). In addition, ATG9-containing vesicles also provide additional lipid/membrane sources for pre-autophagosomal structures [78,79,80]. The elongation and the maturation of autophagosomes require the ATG12 conjugation system which includes ATG5, ATG7, ATG10, ATG12, and ATG16L1 proteins [81,82,83,84]. The covalent conjugation ATG12 and ATG5 is accomplished by two enzymes ATG7 and ATG10. The E1-like enzyme ATG7 activates ATG12 via the formation of a thioester bond between the C-terminus of ATG12 and the cysteine^507^ of ATG7 [85]. Subsequently, activated ATG12 is transferred to the E2 enzyme ATG10 [86], and is eventually conjugated to lys^149^ of ATG5 [81]. The conjugate ATG12-ATG5 subsequently forms a protein complex with ATG16L1 localized on the pre-autophagosomal structure [87]. The ATG12-ATG5-ATG16L complex exhibits E3-like activity that facilitates the conjugation of LC3 to phosphatidylethanolamine (PE). LC3 exists in two forms: the free mature form referred to LC3-I present in the cytosol and the rapidly lipidated form (LC3-II), which is formed during autophagosome formation [88]. LC3 is first processed by autophagin (ATG4) into LC3-I with the exposure of a glycine residue at the C-terminus [89]. The second conjugation system ensures that the cytosolic form of LC3 (LC3-I) links to lipid-conjugated phosphatidylethanolamine (PE) to form LC3-PE (LC3-II). The LC3-II protein is rapidly degraded when the autophagosome fuses with the lysosome. It has also been postulated that WD Repeat Domain, Phosphoinositide Interacting 2 (WIPI2) could act upstream of Atg16L1 to recruit the ATG12-ATG5-ATG16L1 and LC3 proteins to the autophagosome membrane [90]. Since LC3 localizes to the autophagosome membrane, monitoring the free mature form referred to LC3B-I present in the cytosol and the rapidly lipidated form (LCB3-II) by immunoblotting is currently the most widely used method to indirectly assess the levels of autophagosome formation [88]. Importantly, an increase in LC3B-II protein content may not necessarily imply that autophagy is elevated, since insufficient autophagosome fusion with lysosomes or defective lysosome function may also cause higher levels of the LC3B-II protein, irrespective of the degree of autophagy. Currently, autophagic flux assays are the gold standard method for assessing autophagy in a given cell or tissue [91]. In mouse skeletal muscle, autophagy flux has been accomplished by using inhibitors of autophagy, including for instance colchicine [36,92,93,94,95,96], leupeptin [92], and chloroquine [97]. In addition, GFP-LC3 transgenic mice are also often used to evaluate autophagy in vivo [98]. It is worth mentioning that monitoring autophagy in humans in vivo is impossible and that studies in humans are restricted to assessing indirect markers of autophagy activity. Extensive descriptions of autophagy monitoring procedures, as well as their advantages and limitations, have been reviewed in [91,99,100].

It is now generally accepted that autophagy contributes to mammalian cellular homoeostasis by removing damaged organelles under normal physiological conditions and even more under stress conditions, including nutrient and energy deprivation. For example, during acute starvation, autophagic production of amino acids in the liver is important for regulating hepatic gluconeogenesis [101]. There is also evidence that autophagy plays important roles in lifespan and health span in mammals [102,103,104,105]. In mice, muscle-specific deletion of ATG7 (a key regulator of autophagy) was shown to result in severe muscle atrophy, in the deterioration of neuromuscular junctions, and in a reduced lifespan [102]. This severe phenotype is associated with significant mitochondrial abnormalities and multilamellar body structures in mice with muscle-selective inhibition of autophagy [102,106,107,108]. Importantly, plasmid mediated-ATG7 overexpression for 14 days in aged skeletal muscles restored autophagy, ameliorated the integrity of neuromuscular junctions, and increased myofiber size [102]. In humans, mutations in ATG7 cause neurodevelopmental disorders involving neurologic, muscular, and endocrine hypofunction [109]. It was also reported that p62/SQSTM1 (sequestosome-1) accumulates in aged skeletal muscles, suggesting that a decline in autophagy may contribute to sarcopenia [110]. In line with this view, several studies have observed an accumulation of lipofuscin particles in aged muscles from rodents [47] and humans [111,112], suggesting that the autophagy-lysosomal function is defective. Furthermore, inactivation of autophagy regulating genes has also been associated with decreased lifespan in *Caenorhabditis elegans* [113], whereas promotion of autophagy in *Drosophila* extended lifespan [69,114]. It has also been reported that muscle-specific overexpression of FoxO transcription factors in *Drosophila* decreased muscle aging and extended life span, at least in part by sustaining basal levels of autophagy [115]. It is worth mentioning that FoxO transcription factors are now known to be major regulators of several autophagy-related proteins [116,117,118]. Transgenic overexpression of Atg5 (key regulator of autophagy) in mice was shown to improve motor function and extend lifespan by 17%, which is most likely due to enhanced autophagy [119]. There is evidence that aging-related alterations in autophagy likely lead to accumulation of damaged macromolecules and organelles during aging [102,120]. However, it is important to mention here that the molecular machinery regulating autophagy is still only partly understood. Further studies will, therefore, be required to refine our understanding of the exact role that autophagy plays in the skeletal muscle aging process.

## 5. The Role of Mitophagy in Skeletal Muscle Health and Aging

Mitophagy is a specific form of autophagy where damaged mitochondria are selectively eliminated by the autophagy-lysosome system. In healthy skeletal muscles, damaged and depolarized mitochondria are selectively removed through mitophagy pathways (Figure 3). The most extensively studied mitophagy pathway involves the E3 ligase Parkin and PINK1 (Phosphatase and tensin homolog (PTEN)-induced kinase 1) [121,122,123,124,125]. Parkin, an E3 ubiquitin ligase encoded by the *Park2* gene, is a 465 amino acid protein that translocates to depolarized mitochondria to initiate mitophagy [121,122,123,124,125]. Loss of mitochondrial membrane potential (ΔΨm) by mitochondrial uncouplers blocks the import of PINK1 into the inner mitochondrial membrane (IMM). PINK1 then accumulates on the outer mitochondrial membrane (OMM) [122], which, in turn, stimulates the recruitment of Parkin E3 ligase activity from the cytosol followed by removal of damaged mitochondria by autophagosomes [124,125]. Once activated, Parkin has been reported to facilitate ubiquitination of numerous OMM proteins, including Mfn1 [126,127,128], Mfn2 [129], VDAC [130], Tom20 (translocase of the outer membrane 20) [121], and Miro [131]. These ubiquitinated proteins are then degraded by the proteasome and autophagy systems. In healthy mitochondria, cytosolic PINK1 protein is imported into the inner mitochondrial membrane and then degraded by mitochondrial proteases [132,133,134]. Importantly, PINK1 was recently shown to be dispensable for mitochondrial recruitment of Parkin and activation of mitophagy in cardiomyocytes [135]. It should be emphasized here that our current knowledge and understanding of Parkin-dependent mitophagy is largely based on cultured cell experiments in which non-physiological uncouplers such as CCCP were used to induce mitochondrial depolarization.

Mitophagy can also occur through Parkin-independent pathways. For instance, the selective elimination of damaged mitochondria can be achieved by p62/SQSTM1 [130], Bnip3 (BCL2 Interacting Protein 3), and Bnip3L (BCL2 Interacting Protein 3 Like, also called NIX) [136,137,138,139,140]. Moreover, it has been shown that mitochondrial ubiquitin chains can interact with optineurin (*Optn*) and NDP52 (Nuclear dot protein 52 kD) via its LC3 interaction region (LIR) motif to remove dysfunctional mitochondria [141,142]. A recent study also showed that deleting FUNDC1 (FUN14 Domain Containing 1) specifically in skeletal muscles in mice causes mitophagy impairment [143]. As such, when assessing mitophagy, it important to keep in mind that various proteins, such as Bnip3, Bnip3L, NDP52, and p62/SQSTM1, can also trigger mitophagy in a Parkin-independent manner (Figure 3). It should also be mentioned that the translocation of cardiolipin from the inner to outer mitochondrial membrane is another event that can trigger mitophagy [144].

Only few studies have examined the impact of aging on mitophagy pathways in skeletal muscle. Although conflicting data exist [54,145,146,147], several studies reported that mitophagy-related proteins are decreased in the skeletal muscle of old rodents [46,120,148,149,150,151,152]. Similarly, and while not a universal finding [54,145,146,147], Parkin content was reported lower in the skeletal muscle of old mice [152] and rats [151]. Importantly, the majority of studies that reported an increase in Parkin content in aged muscles also reported that mitophagy was impaired/insufficient in aged skeletal muscles [145,146,147]. In humans, we reported that the ratio of Parkin to VDAC (voltage-dependent anion channel) is significantly reduced in atrophied muscles of old men compared to muscles of young men, suggesting that mitophagy might be impaired during muscle aging in humans [9]. In line with this finding, it has been reported that inactive old women have significantly lower expression levels of *Bnip3*, *Beclin 1*, *Atg7*, and *Park2* (the gene coding for Parkin), compared to a more active cohort [150]. These studies suggest the presence of a strong relationship between decreased markers of mitophagy and the development of skeletal muscle atrophy during aging. Based on the available literature, it is possible that disrupted mitophagy and unbalanced mitochondrial dynamics might lead to the accumulation of dysfunctional mitochondria during skeletal muscle aging (Figure 4).

Several recent studies using loss- and gain-of-function approaches in vivo have provided mechanistic insights into the physiological function of Parkin in skeletal muscle health and aging. For instance, it has been reported that muscles from Parkin-deficient mice display lower mitophagy flux in response to acute exercise [147] and endurance training [153]. We and others have reported that muscles of Parkin-deficient mice display several aging-like features, including contractile dysfunction [154] and impaired mitochondrial energetics [147,153,154,155]. We also reported that the loss of Parkin in skeletal muscle triggers an increase in the sensitization of the mPTP [154]. Taken together, these studies highlight the importance of Parkin in the maintenance of mitochondrial integrity in skeletal muscle. Using gain-of-function, we recently showed that Parkin overexpression increased muscle mass, fiber size, and mitochondrial enzyme activity in both young and old muscles [152]. Furthermore, in old mice, we found that Parkin overexpression increased muscle strength, increased mitochondrial content, and protected against oxidative stress, fibrosis, and apoptosis [152]. In line with these findings, Parkin overexpression in skeletal muscles of *Drosophila* led to higher mitochondrial citrate synthase enzymatic activity and attenuated the accumulation of protein aggregates, a marker of cellular aging [156], whereas Parkin loss-of-function in *Drosophila* resulted in a decreased lifespan and increased locomotor defects [157,158]. We also recently showed that Parkin overexpression prevents sepsis-induced skeletal muscle atrophy, likely by improving mitochondrial quality and content [159]. Taken altogether, these studies highlight the importance of proteins regulating mitophagy in skeletal muscle health and aging.

Importantly, mechanistic studies aiming at assessing the role played by proteins involved in Parkin-independent mitophagy in sarcopenia are currently lacking. Assessing whether modulating proteins regulating Parkin-independent mitophagy in aged mice can attenuate sarcopenia might open new therapeutic avenues.

## 6. Enhancing Mitophagy to Counteract Sarcopenia: Potential Applicability and Future Research

In the last years, several studies have provided the proof of principle that stimulating mitophagy can counteract sarcopenia. As discussed above, we have shown that overexpressing Parkin in aged skeletal muscles attenuates sarcopenia [152]. Recently, Dr Auwerx’s research group reported that urolithin A supplementation, a natural dietary microflora-derived metabolite from ingested ellagitannins and ellagic acid, stimulates Parkin-mediated mitophagy and prolongs lifespan in *C. elegans* [160], and improves muscle function and running performance in old rodents while increasing Parkin expression [160]. Administration of urolithin A in the first-in-human clinical trial has also been shown to be safe and to induce a molecular signature of mitochondrial health in the skeletal muscle of healthy sedentary elderly individuals [161]. Another natural compound from tomatoes, tomatidine, has also been shown to attenuate disuse-induced muscle atrophy and improve skeletal muscle mass and function in mice [162], and to prolong lifespan and health span in *C. elegans* by stimulating mitophagy [163]. It was also reported that lifespan extension in response to resveratrol supplementation in rats was associated with an activation of the Pink1/Parkin pathway [164]. It is worth mentioning that two small molecules, T0466 and T0467, were recently shown to increase the mitochondrial translocation of Parkin in human dopaminergic neurons and myoblasts [165]. Interestingly, both compounds were shown to reduce unfolded mitochondrial protein levels, presumably through enhanced PINK1-Parkin signaling [165]. Recently, it was shown that injecting old mice with a miR-181a mimic (miR-181a being a microRNA known to enhance mitophagy in myoblasts) resulted in improved skeletal muscle force, increased myofiber size, and a trend for increased succinate dehydrogenase activity [145]. Taken together, the available literature position mitophagy-modulating compounds as potential therapeutic tools to counteract skeletal muscle atrophy and weakness occurring with aging. Large-scale clinical trials are now needed to assess the efficacy of mitophagy-inducing compounds in fighting sarcopenia.

Interestingly, physical activity, which is well known to improve muscle and mitochondrial health in the elderly [166], was recently shown to stimulate mitophagy [57,153,167,168] and increase Parkin expression in the skeletal muscles of rodents [153] and humans [57,168]. It was also shown that physically active older adults display higher Parkin in mitochondrial fractions prepared from muscle biopsies. It is, therefore, possible that the beneficial effects of physical activity on muscle and mitochondrial health in older individuals [166] might, at least in part, involve an upregulation of mitophagy. Similarly, accumulating experimental evidence indicates that calorie restriction, one of the most efficient interventions to attenuate sarcopenia in rodents [25,26,27] and to improve mitochondrial health [28], also stimulates mitophagy (see [169] for a detailed review). Perhaps not surprisingly, evidence also indicates that fasting likely triggers mitophagy [169]. Interestingly, time-restricted eating, which involves regular periods of fasting, has emerged in the last few years as an intervention with the potential to improve mitochondrial health and increase health span [170,171]. It was even recently suggested that time-restricted eating might be an avenue to fight sarcopenia [172]. Whether time-restricted eating can efficiently stimulate mitophagy in aged skeletal muscles and attenuate the aging-related decline in muscle mass and function represent an interesting research avenue that requires further study.

Recently, it was reported that long-term exposure to a ketogenic diet attenuated the aging-related decline of relative muscle mass [173] and increased markers of mitochondrial content in aged skeletal muscles [174]. Interestingly, the ketone body β-hydroxybutyrate, which is elevated during nutritional ketosis, was shown to induce mitophagy in young and aged myocytes [175] and elicits favorable mitochondrial adaptations in myocytes [176]. These findings combined are particularly interesting when considering the availability of recently developed ketone esters known to significantly increase circulating β-hydroxybutyrate in humans [177]. However, whether a ketogenic diet or ketone supplements can efficiently stimulate mitophagy and attenuate sarcopenia in humans remains unknown.

## 7. Conclusions

In this review, we critically summarized the current state of knowledge on the role played by mitochondrial dynamics and mitophagy in skeletal muscle health and in the development of sarcopenia. Data from mechanistic studies have clearly highlighted that mitochondrial dynamics and mitophagy are critical to skeletal muscle and mitochondrial health. Whether impairment in mitochondrial dynamics contributes to sarcopenia still remains unknown, as the literature is filled with conflicting data. Studies aiming at modulating the expression of proteins regulating dynamics late in life should provide answers as to whether mitochondrial dynamics can be targeted to counter sarcopenia. While conflicting data also exist for mitophagy, a growing body of evidence indicates that mitophagy is impaired in aged skeletal muscles. Importantly, enhancing mitophagy through genetic or nutritional approaches improves skeletal muscle function in aged rodents and improves mitochondrial health in the elderly. Enhancing mitophagy in skeletal muscle, therefore, appears as a promising therapeutic target to prevent or even treat skeletal muscle dysfunction in the elderly.

## Figures and Tables

**Figure 1 ijms-22-08179-f001:**
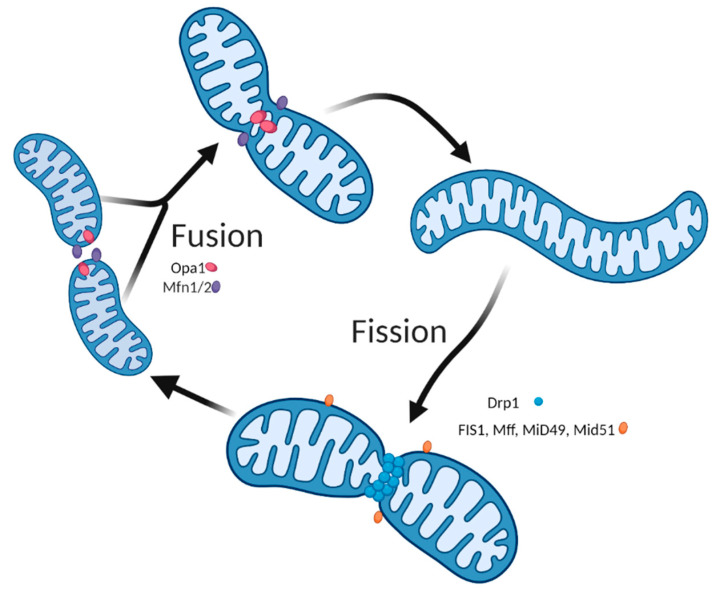
Mitochondrial dynamics. Mitochondria are highly dynamic organelles that can change their morphology through fusion and fission events. Fusion results in mitochondrial elongation (top section), whereas fission fragments mitochondria (bottom section). Mitochondrial fusion is controlled by Mfn1 & Mfn2 on the outer mitochondrial membrane and by Opa1 on the inner mitochondrial membrane. For mitochondrial fission to occur, cytoplasmic Drp1 is recruited on the mitochondrial outer membrane, where it binds to its adaptors, Mff, MiD49, and MiD51, causing a ring- or spiral-like structure around mitochondrial constriction sites. Created with BioRender.com.

**Figure 2 ijms-22-08179-f002:**
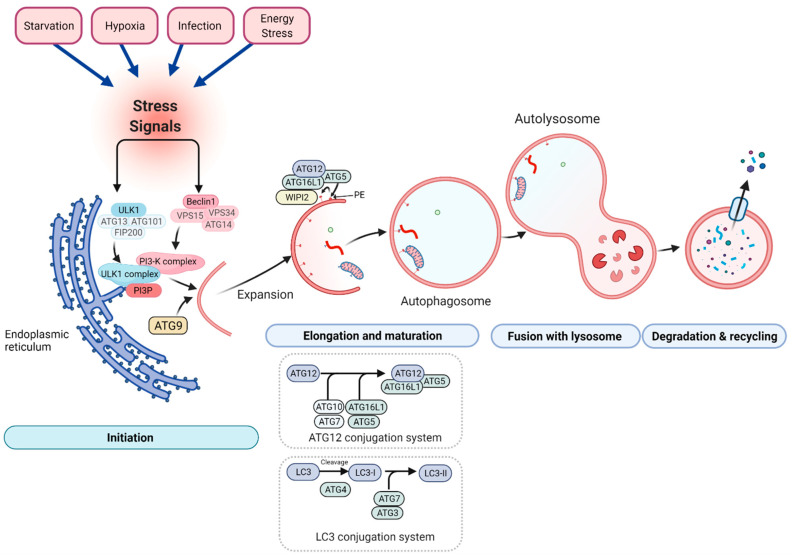
Schematic representation of the autophagy machinery. Autophagy consists of several stages: initiation, elongation and maturation, fusion to lysosome, and degradation and recycling. Autophagy is initiated by stress signals (top section). Both the Unc-51 Like Autophagy Activating Kinase (ULK1) and phosphatidylinositol-3 kinase (PI3-K) complexes are essential for initiation of the phagophore. ATG9 is involved in the delivery of lipids for the phagophore’s membrane expansion. The elongation and the maturation of the autophagosomal membrane are dependent on two ubiquitin-like conjugation systems, the ATG12 and microtubule-associated protein light chain 3 (LC3) conjugation systems. The first conjugation system requires ATG12, ATG5, ATG7, ATG10, and ATG16L1 proteins. This system works as E3-like enzymes for the second LC3 conjugation system. The LC3 conjugation system requires ATG4, ATG7, and ATG3 enzymes. Both conjugation systems are essential for the elongation and maturation of the autophagosome. The WD Repeat Domain, Phosphoinositide Interacting 2 (WIPI2) protein also promotes conjugation of LC3 with ATG16L1. Created with BioRender.com.

**Figure 3 ijms-22-08179-f003:**
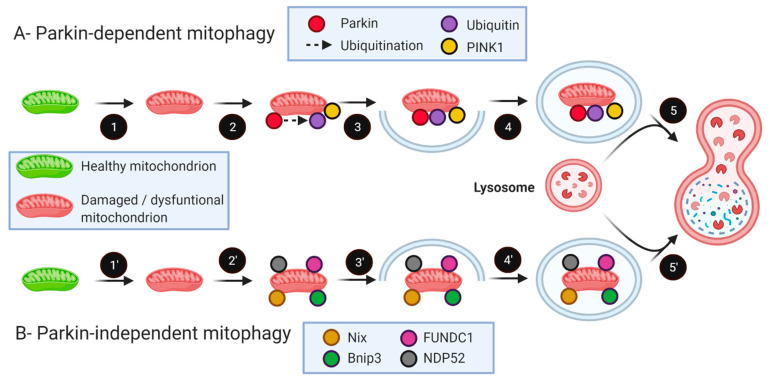
Parkin-dependent and Parkin-independent mitophagy pathways. (**A**) Parkin-dependent mitophagy pathway: in dysfunctional mitochondria, PINK1 stabilizes on the OMM leading to the recruitment of Parkin through a series of modifications, including ubiquitination and phosphorylation of Parkin and ubiquitin. During mitophagy, Parkin polyubiquitinates various OMM proteins, including mitofusins (Mfn1 and Mfn2), VDAC, Tom20, and Miro. Ubiquitinated proteins are recognized by several adaptors, leading to the recruitment of an autophagosome and subsequent degradation of the mitochondrion. (**B**) Parkin-independent mitophagy pathway: mitochondrial proteins, such as Nix, Bnip3, FUNDC1, and NDP52, can act as mitophagy receptors, promoting the recruitment of the autophagosome around the cargo. Created with BioRender.com.

**Figure 4 ijms-22-08179-f004:**
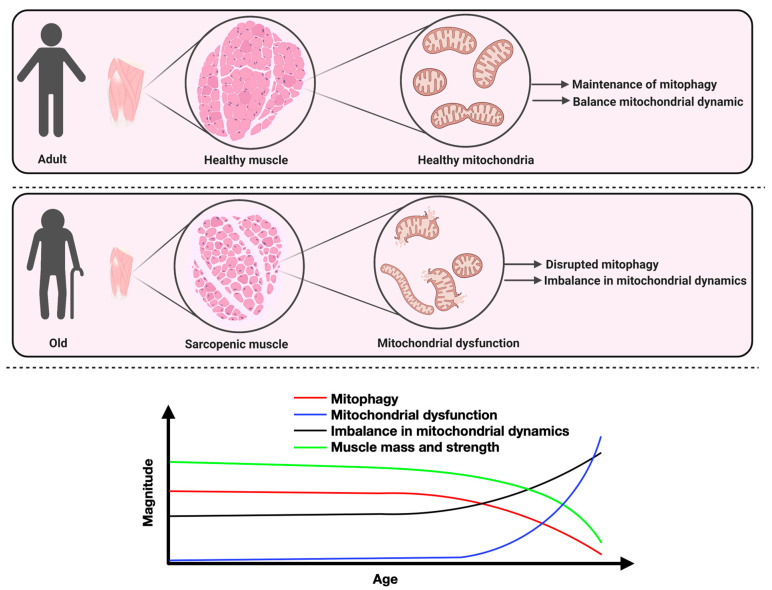
Conceptual model linking impairments in mitochondrial dynamics and mitophagy to the accumulation of mitochondria and the loss of muscle mass and function during aging. In healthy skeletal muscles, mitochondrial dynamics and mitophagy play central roles in maintaining mitochondrial integrity. The mitochondrial dynamics balance and mitophagy levels are represented by a black and red line, respectively; the blue line represents the level of mitochondrial dysfunction; the green line represents muscle mass and strength. Created with BioRender.com.

**Table 1 ijms-22-08179-t001:** Compilation of studies that have investigated the impact of aging on the expression or content of proteins regulating mitochondrial dynamics.

Species	Sex	Tissue	Model	Main Results	Ref.
Mice	Male & female (combined)	QUAD	Youths/Middle-aged 3–6 vs. 8–15 months old	Mfn1 & Mfn2: ↑ Opa1 & Drp1: NS Fis1: ↓	[46]
Rats (Fischer 344 Brown Norway)	Male	EDL	Youths/Aged 5 vs. 35 months old	Mfn2, Fis1 & Opa1: ↑ Drp1: NS	[47]
Rats (Wistar)	Not specified	GAS	Youths/Aged 3 vs. 26 months old	Fis1: ↑	[48]
Rats (Fischer 344 Brown Norway)	Male	TA	Youths/Aged 5 vs. 35 months old	Fis1 & Drp1;:↑ Mfn2: ↓ Opa1: NS	[49]
Rats (Sprague–Dawley)	Male	GAS & SOL	Youths/Aged 9 vs. 22 months old	Drp1 (SOL & GAS):↑ Mfn2 & Fis1 (GAS): ↑	[50]
Rats (Wistar)	Male	GAS	Youths/Aged 3 vs. 26 months old	Fis1 & Mfn1: ↑	[51]
Mice	Male	GAS	Youths/Aged 2–3 vs. 22–24 months old	Mfn2/Drp1 ratio:↑ Opa1, Drp1, Mfn1 & Mfn2: NS	[52]
Rats (Sprague-Dawley)	Male	GAS & TRI	Youths/Aged 3 vs. 22 months old	Opa1 & Mfn1 (GAS & TRI): ↑ Fis1 (GAS): ↓ Fis1 (TRI): ↑	[53]
Mice	Not specified	TA	Youths/Aged 6 vs. 18 months old	OPA1: ↓	[36]
Mice	Not specified	GAS	Youths/Aged 6 vs. 22 months old	Mfn1, Mfn2, Opa1 & Fis1: ↓ Drp1: NS	[34]
Mice (C57BL/6J)	Female	TA	Youths/Aged 2 vs. 24 months old	Fis1 & Mfn2: ↑ Drp1 & Opa1: NS	[54]
Rats (Sprague-Dawley)	Male	Muscles	Youths/Aged 5 vs. 25 months old	Drp1: ↑ Opa1: NS Mfn2 & Fis1: ↓	[55]
Humans	Male	VL	Younger men (20 ± 1 y) vs. older men (74 ± 3 y)	Mfn1, Mfn2 & Fis1: NS	[56]
Humans	Male	VL	Younger men (22 ± 1 y) vs. older men (67 ± 2 y)	Opa1, Mfn2, Fis1: NS	[57]
Humans	Male & female (combined)	VL	Younger (24 ± 3 y) vs. older adults (78 ± 5 y)	Opa1, Mfn2, Fis1 & Drp1: NS	[58]
Humans	Male & female (combined)	VL	Younger (23 ± 1 y) vs. older adults (75 ± 1 y)	Mfn2, Fis1, Drp1: NS Opa1: ↓	[59]
Humans	Male & female (combined)	VL	Younger (≈23 y) vs. older adults (≈70 y)	Mfn2: ↓ Drp1: trend for ↓	[60]

Abbreviations: Quadriceps (QUAD); Tibial anterior (TA); Soleus (SOL); Extensor digitorum longus (EDL); Vastus lateralis (VL) and Gastrocnemius (GAS); Triceps (TRI); ↓ indicate decreased levels, whereas ↑ indicate increased levels in aged skeletal muscles. No significant change (NS).

## Data Availability

Not applicable.

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
