# Peer review of "Mitochondrial Dynamics and Mitophagy in Skeletal Muscle Health and Aging"

_ijms, 2021, doi:10.3390/ijms22158179_

Round 1

Reviewer 1 Report

I think this is well written and organized.

I would invite the authors to expand the section of possible countermeasures, proposing different training schemes.

I would include a section on the ketogenic diet and/or intermittent fasting, which have been shown to be effective in improving mitochondrial health.

We could also insert a part concerning  miRNAs as there is some work right on the sarcopenia-mitochondria link

Author Response

We thank the reviewer for his/her positive comments on our manuscript. As requested, the section on possible countermeasures has been expanded.

  • The following text on calorie restriction and time restricted feeding was added on p12:

Similarly, accumulating experimental evidence indicates that calorie-restriction, one of the most efficient interventions to attenuate sarcopenia in rodents [25-27] and to improve mitochondrial health [28], also stimulates mitophagy (see [171]for a detailed review). Perhaps not surprisingly, evidence also indicate that fasting likely triggers mitophagy [171]. Interestingly, time restricted eating, which involves regular periods of fasting, has emerged in the last few years as an intervention with the potential to improve mitochondrial health and increase health span [172,173]. It was even recently suggested that time restricted eating might be an avenue to fight sarcopenia [174]. Whether time restricted eating can efficiently stimulate mitophagy in aged skeletal muscles and attenuate the aging-related decline in muscle mass and function requires further studies.”

  • The following text on miRNA was added on page 12:

Recently, it was shown that injecting old mice with a miR-181a mimic (miR-181a being a microRNA known to enhance mitophagy in myoblasts) resulted in improved skeletal muscle force, increased myofiber size and a trend for an increase in succinate dehydrogenase activity [146]. »

  • The following text on ketogenic diet and ketone supplements was added on page 13:

Recently, it was reported that long term exposure to a ketogenic diet attenuated the aging related decline in relative muscle mass [175] and increased markers of mitochondrial content in aged skeletal muscles [176]. Interestingly, the ketone body β-hydroxybutyrate, which is elevated during nutritional ketosis, was shown to induce mitophagy in young and aged myocytes [177] and elicit favorable mitochondrial adaptations in myocytes [178]. These findings combined are particularly interesting when considering the availability of recently developed ketone esters known to significantly increase circulating β-hydroxybutyrate in humans [179]. However, whether a ketogenic diet or ketone supplements can efficiently stimulate mitophagy and attenuate sarcopenia in humans remains unknown.”

Reviewer 2 Report

This manuscript is a comprehensive review which focuses on a clinically important issue. The relevant findings are discussed in a logical order. The authors should check the grammar and spelling throughout the manuscript (e.g. lanes 45, 88, 107, 182, 184, 187, 189, 223, 224, 229, 241, 265, 280, 299, 329, 380, 389, 390, 421). The statement in lanes 164-165 requires citation(s). Furthermore, all abbreviations should be given when first mentioned in the text.

Author Response

We thank the reviewer for his/her positive comments on our manuscript. We have thoroughly checked our manuscript for grammatical and spelling errors. We have also checked that all abbreviations are given when first mentioned in the text.